# Layilin regulates Treg motility and suppressive capacity in skin

Victoire Gouirand[1], Sean Clancy[1], Courtney Macon[1], Jose Valle[2], Mariela Pauli[2], Hong-An Truong[2], Jarish Cohen[1], Maxime Kinet[1], Margaret M Lowe[1], Samuel J Lord[3], Kristen Skruber[3], Hobart Harris[4], Esther Kim[4], Isaac Neuhaus[1], Karin Reif[2], Ali A Zarrin[2], R Dyche Mullins[3], Michael D Rosenblum[1]*

[1]Department of Dermatology, University of California, San Francisco, San Francisco, United States; [2]TRex Bio Inc, South San Francisco, United States; [3]Department of Cellular and Molecular Pharmacology, University of California, San Francisco, San Francisco, United States; [4]Department of Surgery, University of California, San Francisco, San Francisco, United States

## eLife Assessment

This study reports **valuable** findings on the role of layilin in the motility and suppressive capacity of clonal expanded regulatory T cells (Tregs) in the skin. Although the strength of the study is utilizing conditional knock-out mice and human skin samples, the analysis of the molecular mechanism by which layilin affects Treg function is **incomplete**. The study will be of interest to medical scientists working on skin immunology.

**Abstract** Regulatory T cells (Tregs) are essential for maintaining immune tolerance in both lymphoid and non-lymphoid tissues. We discovered that layilin, a C-type lectin receptor, is predominantly expressed on Tregs in skin. Layilin was highly expressed on a subset of clonally expanded 'effector' Tregs in both healthy and psoriatic human skin. Layilin-expressing Tregs exhibited a transcriptional profile indicative of enhanced adhesion. Deletion of layilin in Tregs in mice in vivo resulted in significantly attenuated skin inflammation. Mechanistically, layilin enhanced in vitro human Treg adhesion via modulation of lymphocyte function-associated antigen-1, resulting in distinct cytoskeletal alterations consistent with enhanced focal adhesion and lamellipodia formation. Taken together, we define layilin as a critical regulator of Treg-suppressive capacity by modulating motility and adhesion in a non-lymphoid tissue.

*For correspondence:
michael.rosenblum@ucsf.edu

## Introduction

Regulatory T cells (Tregs) play a crucial role in mediating peripheral immune tolerance. Although present in lymphoid organs, specialized subsets of these cells stably reside in non-lymphoid tissues, including skin, gut, and lung (*Muñoz-Rojas and Mathis, 2021*; *Ali and Rosenblum, 2017*). Subsets of Tregs in skin play a crucial role in suppressing hair follicle autoreactivity (*Cohen et al., 2024*). In addition, these cells facilitate hair follicle cycling (*Ali et al., 2017*), augment wound healing (*Nosbaum et al., 2016*), and establish and maintain tolerance to skin commensals (*Scharschmidt et al., 2015*; *Scharschmidt et al., 2017*). Many autoimmune and chronic inflammatory diseases are thought to result from an imbalance in the relative abundance and functional state of proinflammatory and regulatory cells. Thus, elucidating the fundamental mechanisms these cells utilize in both healthy and inflamed tissues is crucial in attempts to develop novel therapies aimed at restoring tissue immune homeostasis.

Layilin is a cell surface C-type lectin receptor (*Borowsky and Hynes, 1998*; *Bono et al., 2001*). This family of receptors plays a role in many cellular functions, including adhesion and cell signaling (*Brown et al., 2018*). Previously, we and others have shown that layilin is preferentially expressed on exhausted CD8+ T cells infiltrating human and murine tumors (*Zheng et al., 2017*; *Fu et al., 2014*; *Mahuron et al., 2020*). In addition, intratumoral Tregs express high levels of layilin (*Bhairavabhotla et al., 2016*; *Mehta et al., 2021*; *De Simone et al., 2016*). Mechanistically, we have shown that layilin functions to anchor Tregs in non-lymphoid tissues, and in doing so, limits their suppressive capacity (*Mehta et al., 2021*). On CD8+ T cells, layilin co-localizes with integrin αLβ2 (lymphocyte function-associated antigen-1 [LFA-1]), enhancing its adhesive properties (*Mahuron et al., 2020*). Despite emerging evidence that layilin plays an important role in the biology of Tregs in non-lymphoid tissues, little is known about the potential role it plays in autoimmune disease. In addition, whether layilin functions to modulate LFA-1 activation on Tregs remains to be determined. Here, we interrogate the functional biology of Treg expression of layilin in inflammatory skin disease.

## Results

### Layilin is expressed on activated 'effector' Tregs in human inflammatory skin disease

To begin to understand the expression pattern of layilin on immune cells in chronic inflammatory skin diseases, we performed single-cell RNA sequencing of CD45+ cells sorted from healthy or psoriatic skin. Consistent with previous observations (*Bhairavabhotla et al., 2016*), the main population expressing *LAYILIN* are Tregs (*Figure 1A–C*). There were no changes in the relative expression of *LAYN* per cell when comparing Treg clusters from psoriasis skin with healthy control skin (*Figure 1— figure supplement 1A*), suggesting that *LAYN* expression is not increased on Tregs in inflamed skin. Because layilin is expressed on ~40–50% of Tregs in both healthy and inflamed skin (*Mehta et al., 2021*), we sought to understand how layilin-expressing Tregs differed from layilin-non-expressing Tregs at the transcriptional level. To this end, we subclustered Tregs from psoriatic skin and performed pseudo-bulk analysis based on *LAYN* gene counts (threshold greater than 0 considered positive for *LAYN* expression) (*Figure 1D and E*). We found that *LAYN*-expressing (*LAYN>0*) and non-expressing (*LAYN<0*) cells are relatively similar to equivalent levels of *FOXP3* and *CTLA-4* expression (*Figure 1F*). Similarly to the observation made in *Figure 1—figure supplement 1A*, we noticed most differences are driven by *LAYN* expression (*Figure 1—figure supplement 1B*). Focusing on psoriatic skin, we found that 636 genes were significantly differentially expressed (DEGs) between *LAYN*-positive and *LAYN*-negative Tregs (*Figure 1G*). Interestingly, several *TNFRSF* family genes and several adhesion molecules, including integrin coding genes (*ITGB7* and *ITGB6*) or cell–cell adhesion (*CADM1*), were expressed at significantly higher levels in *LAYN*-expressing Tregs (*Figure 1G*). Gene set enrichment analysis (GSEA) on DEGs revealed pathways for T cell activation and cell migration and adhesion to be enriched in *LAYN*-expressing Tregs (*Figure 1H*).

T cell receptor (TCR) engagement is a potent inducer of layilin expression on Tregs (*Mehta et al., 2021*). Thus, we hypothesized that layilin-expressing Tregs may be specific for tissue antigens and consequently more clonal than Tregs in skin with undetectable layilin expression. To test this hypothesis, we performed TCR repertoire analysis using the same subclustering of Tregs as in the pseudo-bulk analysis (*Figure 1I and J*). Interestingly, *LAYN*-expressing Tregs had more expanded clones, with the top 15 expressed clonotypes being more abundant in *LAYN*-expressing cells. These findings are similar to previous observations made in CD8+ T cells (*Mahuron et al., 2020*). Taken together, transcriptional immunophenotyping of Tregs isolated from psoriatic skin suggests that *LAYN* expression correlates with more activated, more motile, and clonally expanded cells.

### Layilin has a minimal effect on Treg activation and suppressive capacity in vitro

Activated lymphocytes express receptors and pathways that can either promote or attenuate their effector functions (*Carlino et al., 2021*). In addition, ex vivo activation of lymphocytes can induce molecular pathways that only play a meaningful biologic role when the cells are in their natural in vivo setting. Thus, we sought to determine whether layilin expression on Tregs plays a role in promoting their activation state or suppressive capacity in vitro. To do so, we isolated Tregs from peripheral

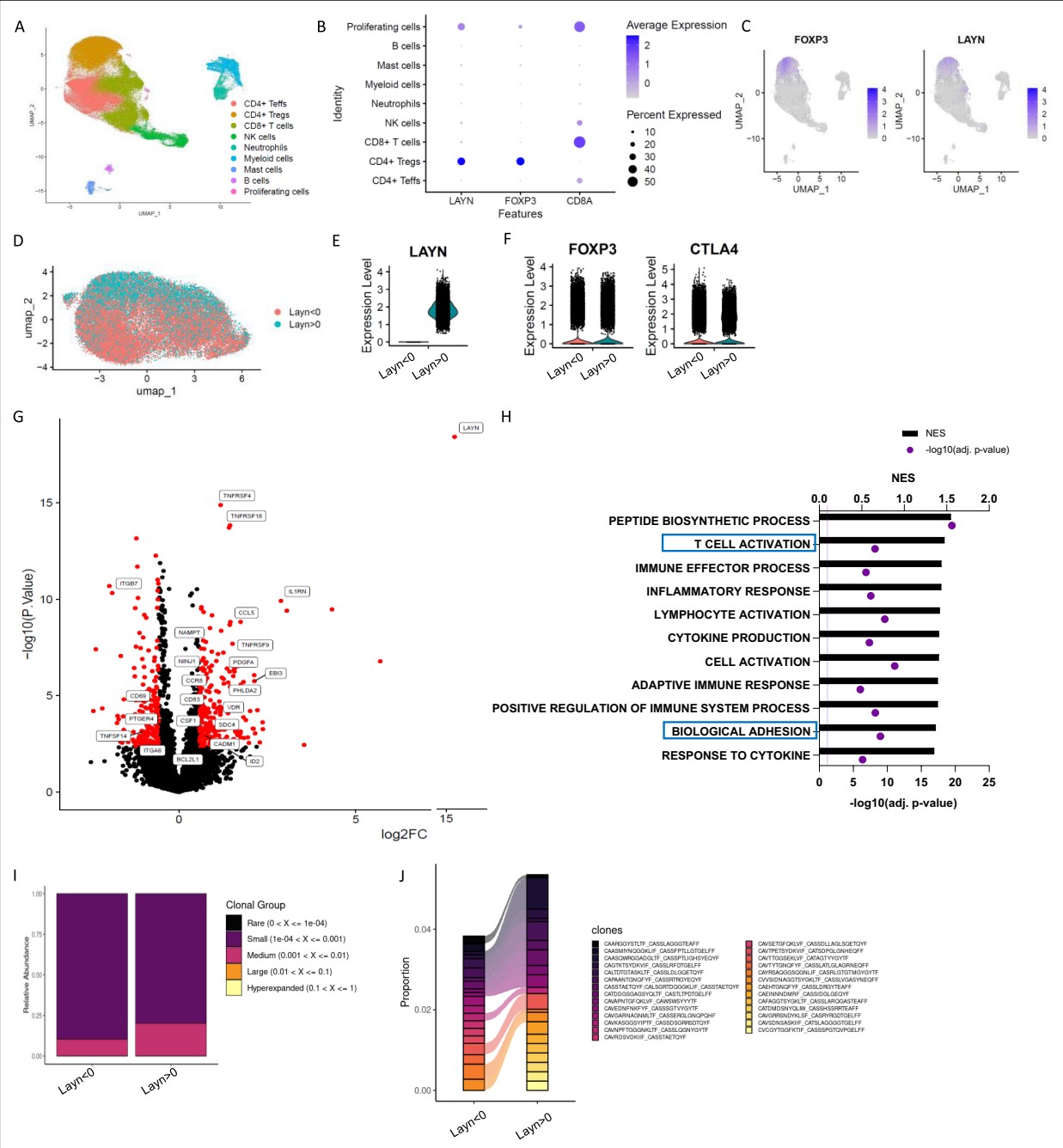

**Figure 1.** Layilin is preferentially expressed on regulatory T cells (Tregs) in skin and correlates with activation and motility signatures. (**A–C**) Single-cell RNA-seq of FACS-purified CD45⁺ cells from healthy and psoriatic skin. n=5 healthy skins and 5 psoriatic skins. (**A**) Representative Uniform Manifold Approximation and Projection (UMAP) showing all clusters found. (**B**) Dot plot of *LAYN, FOXP3,* and *CD8A* expression by clusters. (**C**) UMAP of *FOXP3* and *LAYN* expression in skin. (**D**) UMAP of Tregs subset showing in red *LAYN* non-expressing Tregs (LAYN<0) and blue *LAYN*-expressing Tregs (LAYN>0) based on gene count >0. (**E, F**) Violin plot of *LAYN* (**E**), *FOXP3,* and *CTLA4* (**F**) in subset Tregs from skin followed by resclustering based on *LAYN* gene count >0. (**G**) Volcano plot of *LAYN*-expressing Tregs (right) compared to *Layn*-non-expressing Tregs (left) obtained from pseudo-bulk analysis performed on Tregs subset. (**H**) Gene set enrichment analysis of top enriched pathways in *LAYN*-expressing Tregs compared to *LAYN*-non-expressing Tregs associated with panel (**F**). (**I, J**) T cell receptor (TCR) analysis run on Treg cluster subset by *LAYN* expression. In (**I**), we show the relative abundance of TCR clonotype frequencies separated into five groups: rare being expressed between 0 and 10⁴, small between 10⁴ and 0.001, medium between 0.001 and 0.01, large between 0.01 and 0.1, and hyperexpanded between 0.1 and 1. (**J**) Representation of the top expressed clonotypes in a subset of Tregs from skin resclustered according to *LAYN* gene count >0.

The online version of this article includes the following figure supplement(s) for figure 1:

**Figure supplement 1.** Layilin Expression on Tregs.

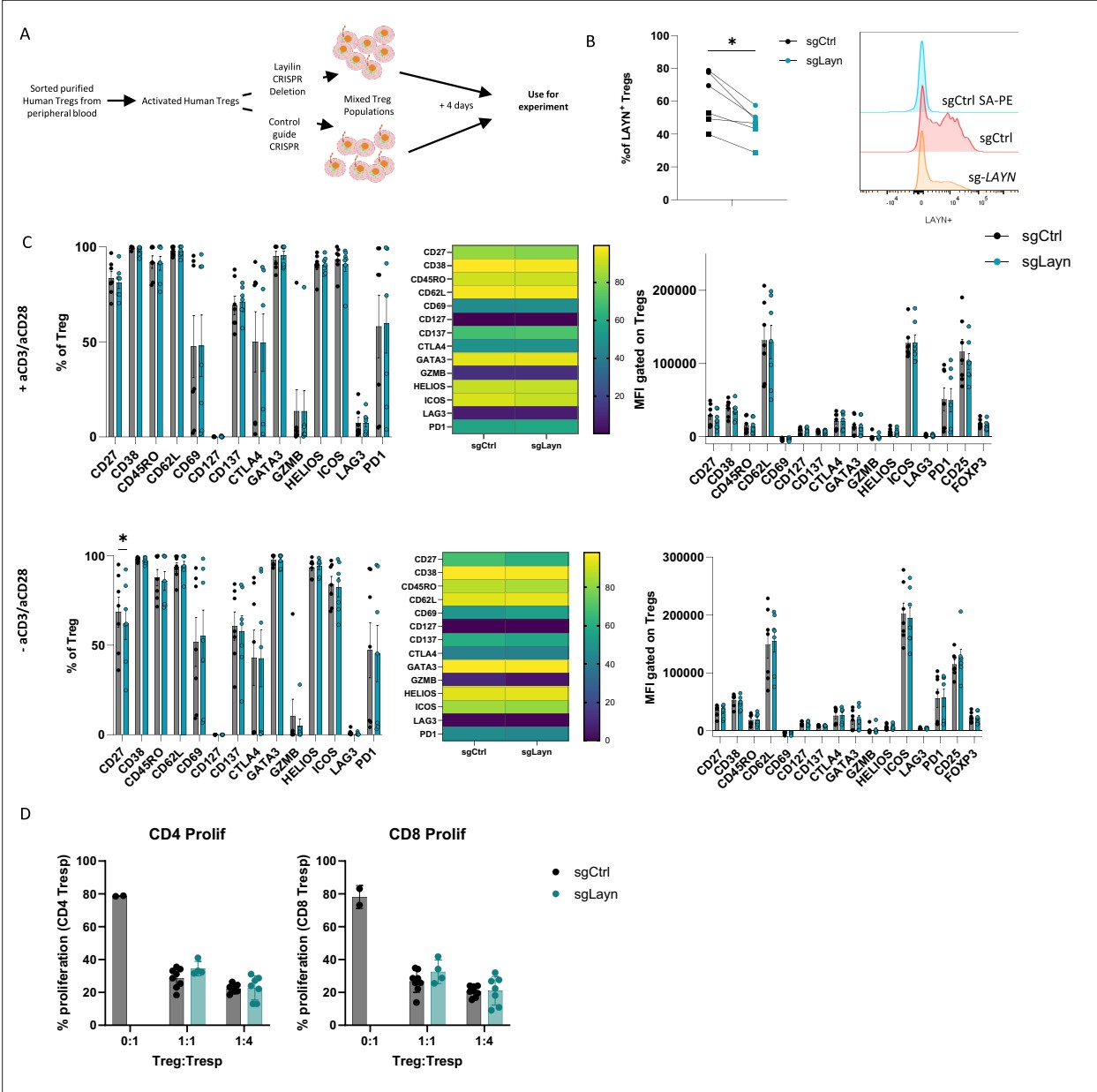

**Figure 2.** Layilin does not affect regulatory T cell (Treg) activation or suppression in vitro. (**A**) Schematic experimental design strategy starting with the isolation of human Tregs from peripheral blood, activated and expanded using anti-CD3/anti-CD28 stimulation and IL-2 for 12 days. We deleted *LAYN* using CRISPR/Cas9, and after 4 days, cells were used for the experiment. (**B**) Frequency of LAYN+ Tregs gated on live CD8-CD4+CD25+FOXP3+ cells and representative expression of layilin. n=6 donors from two independent experiments. Significance was defined by a two-tailed Student's *t*-test. *p<0.05. (**C**) Frequencies (on the left) and mean fluorescence intensity (MFI, on the right) of sg-*LAYN* and sg-CTRL Tregs gated on live CD8-CD4+CD25+FOXP3+ cells expressing the different activation Treg markers listed on the x-axis. The top-row graphs show results from cells stimulated with anti-CD3/anti-CD28; the bottom graphs show results obtained from cells without anti-CD3/anti-CD28 stimulation. n=6 donors from two independent experiments. Significance was defined by using a multiple-paired *t*-test. (**D**) Proliferation percentage measured using cell trace violet for CD4+ T responder (left panel) or CD8+ T responder (right panel) after 3 days of suppression assay. Results are presented in the x-axis, and the different ratios of sg-*LAYN* and sg-CTRL Tregs and T responders used for the assay represent two independent experiments. Significance was defined by two-way ANOVA, followed by Tukey's multiple-comparisons test.

blood of healthy volunteers and expanded them ex vivo with anti-CD3 and anti-CD28-coated beads and high-dose IL-2. Expanded cells were specifically edited for *LAYN* using a well-established CRISPR-Cas9 electroporation protocol (*Albanese et al., 2022*; *Figure 2A*). Four days post-electroporation, we consistently observed a maximal reduction in *LAYN* expression by ~30–50% compared to single

guide-treated control cells (*Figure 2B*). After 4 days of suboptimal activation post-electroporation, cells were rested and cultured with or without TCR restimulation. Subsequently, the cells underwent spectral flow cytometry analysis using a detailed panel of Treg activation markers. In these experiments, deletion of layilin did not influence Treg activation across multiple donors with or without TCR stimulation (*Figure 2C*). To determine if deleting layilin on Tregs impairs their suppressive function in vitro, we performed standard Treg suppression assays with cells edited for *LAYN* or control guide RNAs. In these experiments, deleting layilin on 30–50% of Tregs had no effect on their ability to suppress either CD4+ or CD8+ T cell proliferation (*Figure 2D*). Taken together, this data suggests that layilin plays a minimal role, if any, in Treg activation and suppressive capacity in vitro.

## Layilin attenuates Treg suppression in vivo

We have previously shown that layilin acts to anchor Tregs in tissues and that deletion selectively in these cells results in enhanced tumor growth (*Mehta et al., 2021*). Thus, despite having limited to no role in Treg activation and suppression in vitro (*Figure 2*), we hypothesized that layilin expression on Tregs in vivo would attenuate their suppressive capacity. To test this hypothesis, we utilized the well-established imiquimod (IMQ) model of skin inflammation (*Gangwar et al., 2022*; *van der Fits et al., 2009*), where Tregs are known to play a significant role (*Choi et al., 2020*; *Chen et al., 2022*; *Hartwig et al., 2018*). To inducibly and selectively delete *Layn* in Tregs, we crossed *Foxp3$^{CreERT2}$* mice (*Foxp3$^{Cre}$*) (*Rubtsov et al., 2010*) with *Layn$^{fl/fl}$* mice (generated previously by our lab; *Mahuron et al., 2020*). To delete layilin in Tregs, the resultant *Foxp3$^{Cre}$/Layn$^{fl/fl}$* mice were treated with systemic tamoxifen from 6 days to 2 days prior to beginning IMQ application on dorsal back skin for 6 days (*Figure 3A*). Because there are no working antibodies that reliably detect murine layilin by flow cytometry, we performed qRT-PCR on sort-purified Tregs from *Foxp3$^{Cre}$/Layn$^{fl/fl}$* and *Layn$^{fl/fl}$* mice after tamoxifen treatment to assay for *Layn* deletion. These experiments confirmed a significant reduction in *Layn* expression in Tregs from *Foxp3$^{Cre}$/Layn$^{fl/fl}$* mice (*Figure 3—figure supplement 1A*). Clinical signs of disease, including skin lesions, scaling, and erythema, were quantified in a blinded fashion throughout the duration of the IMQ treatment. Consistent with our hypothesis, we observed a significant reduction in the clinical severity of skin inflammation in *Foxp3$^{Cre}$/Layn$^{fl/fl}$* mice treated with tamoxifen compared to age- and gender-matched littermate *Foxp3$^{Cre}$* control mice treated with tamoxifen (*Figure 3B and C*). Flow cytometric quantification of immune cell infiltrate and cellular cytokine production was performed on day 6. *Foxp3$^{Cre}$/Layn$^{fl/fl}$* mice had a significant reduction in the absolute number of skin infiltrating CD4+ Teff cells and CD8+ T cells compared to controls (*Figure 3D*). In addition, there was a significant reduction in the number of TNFa-producing CD4+ Teff cells, IL-17-producing CD8+ T cells, and IL-17-producing Tregs in *Foxp3$^{Cre}$/Layn$^{fl/fl}$* mice compared to controls (*Figure 3E*, *Figure 3—figure supplement 1B*). There were no significant differences in the expression of select Treg activation markers between layilin-deleted and control Tregs (*Figure 3F*, *Figure 3—figure supplement 1C*).

To comprehensively interrogate how layilin deletion alters Treg biology in vivo at the transcriptional level, we performed bulk RNA sequencing. RNA was isolated from GFP-expressing Tregs sort purified from back skin after 4 days of IMQ treatment and subjected to low-input RNA sequencing (*Figure 3—figure supplement 1D and E*). These experiments identified several DEGs in *Foxp3$^{Cre}$/Layn$^{fl/fl}$* mice relative to *Foxp3$^{Cre}$* control mice both treated with tamoxifen (*Figure 3G*). Interestingly, GSEA of DEGs revealed enrichment in pathways involved in cellular movement, locomotion, and migration (*Figure 3H*). Taken together, these results reveal that layilin attenuates Treg suppression in vivo without altering their activation capacity and is consistent with previous studies that suggest that the anchoring function of layilin limits Treg's ability to optimally mediate immune regulation in skin (*Mehta et al., 2021*).

## Layilin enhanced Treg adhesion in an LFA-1-dependent manner

We have previously shown that layilin co-localizes with LFA-1 on CD8+ T cells and functions to augment LFA-1-mediated adhesion of these cells (*Mahuron et al., 2020*). Thus, we set out to determine whether layilin functions in a similar fashion on Tregs. In addition, we sought to determine whether layilin expression on Tregs influences motility pathways at the transcriptional level. To this end, we sorted purified Tregs from human peripheral blood, CRISPR-edited these cells for *Layn* (or control guide RNA), sort-purified these cells based on layilin expression, and performed bulk RNA sequencing on highly pure layilin-expressing or layilin-edited cells (*Figure 4A*). Consistent with the experiments

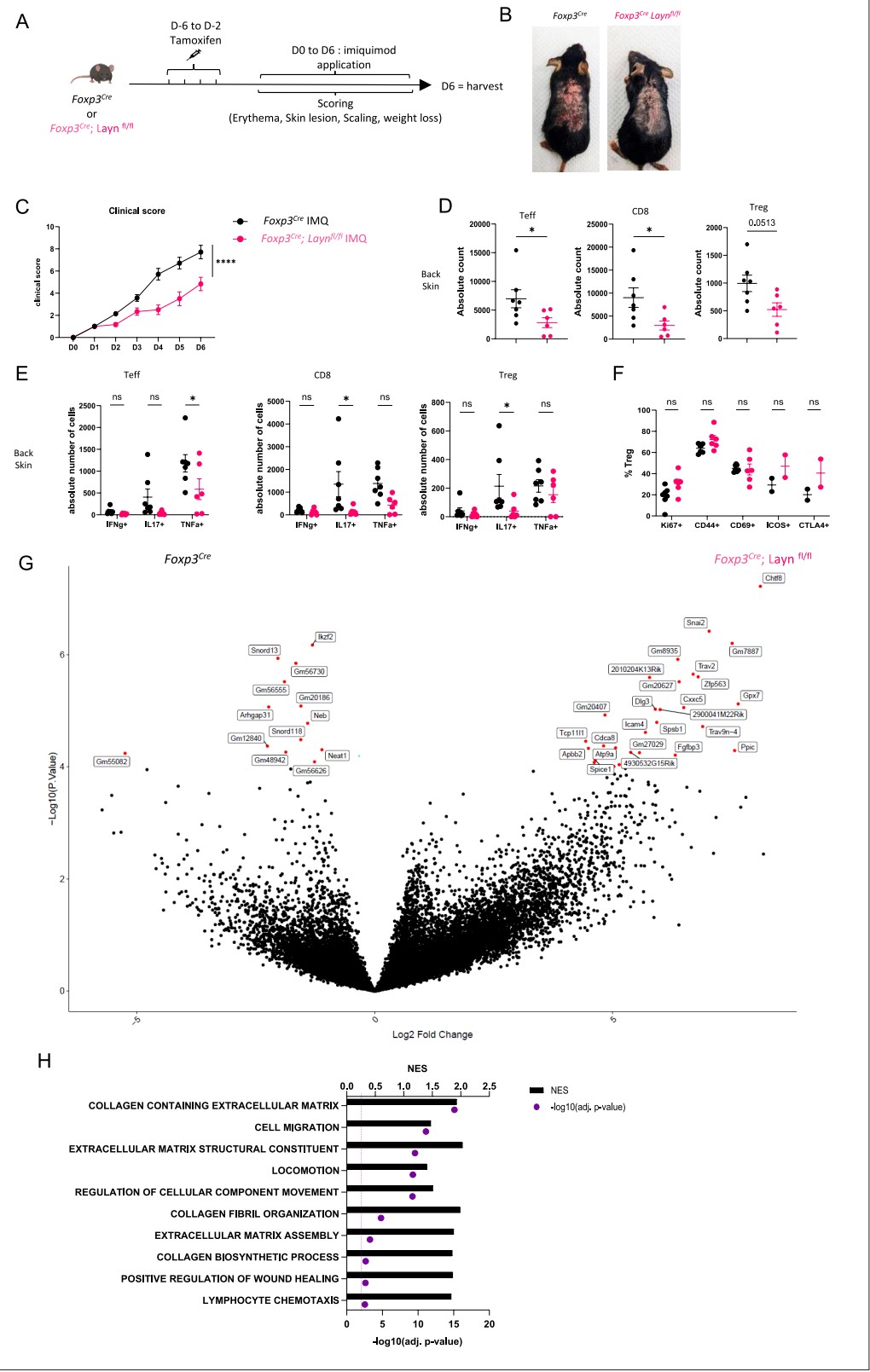

**Figure 3.** Layilin limits regulatory T cell (Treg)-mediated suppression of inflammation in vivo. (**A**) Experimental design strategy of the imiquimod (IMQ) model. After 4 days of tamoxifen injection, mice were rested for 2 days before the start of the IMQ treatment. They received a daily dose of IMQ on shaved back skin and were scored for erythema, scaling, skin lesions, and weight loss. After 6 days of treatment, the mice were euthanized.

*Figure 3 continued on next page*

*Figure 3 continued*

(**B**) Representative pictures of mouse back skin after 5 days of treatment from each experimental group.
(**C**) Overall clinical score over time. n=7 mice from three independent experiments. Significance was defined by two-way ANOVA. ****p<0.0001. (**D**) Absolute count of Teff, CD8, and Tregs obtained from each experimental group, respectively, gated on live $CD3^+TCR\beta^+$ and $CD4^+FOXP3^-$ or $CD8^+$, $CD4^+FOXP3^+$. n=7 and 6 mice per group from three independent experiments. Significance was defined by a two-tailed Student's *t*-test. *p<0.05.
(**E**) Absolute count of cells Teff, CD8, and Treg either $IFN\gamma+$, IL17+, or $TNF\alpha^+$. n=7 and 6 mice per group from three independent experiments. Significance was defined by a multiple-paired *t*-test. *p<0.05. (**F**) Frequencies of Tregs activation markers gated on live $CD3^+TCR\beta^+$ $CD4^+FOXP3^+$ Tregs. n=7 and 6 mice per group from three independent experiments, except ICOS and CTLA from two mice and one experiment. Significance was defined by a multiple-paired *t*-test. (**G**) Volcano plot of differential expression of genes from sorted Tregs gated on $GFP^+$ from *Foxp3^Cre^; Layn^fl/fl^* (on the right) or *Foxp3^Cre^* (on the left) after 4 days of treatment. Red dots represent differentially expressed genes, and black dots are not differentially expressed ones. n=5 mice per group from two independent experiments. (**H**) Top deregulated pathways obtained from a gene set enrichment analysis comparing *Foxp3^Cre^; Layn^fl/fl^* to *Foxp3^Cre^* Tregs. The top axis and bar plot represent the normalized enrichment score (NES). The bottom axis and dot represent the adjusted p-value associated with pathways enrichment. n=5 mice per group from two independent experiments.

The online version of this article includes the following figure supplement(s) for figure 3:

**Figure supplement 1.** Immunophenotyping of imiquimod experiments in mice with Layilin-defieicnt Tregs.

described above where layilin was deleted in Tregs in vivo in mice (*Figure 3*), several genes involved in cellular locomotion, migration, and adhesion were differentially expressed in layilin-expressing vs. layilin-edited cells (*Figure 4B and C*).

LFA-1 is critical for the development and function of Tregs (*Marski et al., 2005*; *Wohler et al., 2009*). LFA-1 is present either in a low-affinity inactive form or a high-affinity active form (*Sun et al., 2019*; *Kondo et al., 2022*). In its active form, LFA-1 mediates binding to intercellular adhesion molecule 1 (ICAM-1) and cell–cell adhesion (*Abram and Lowell, 2009*; *Walling and Kim, 2018*). To test if layilin modulates LFA-1 function on Tregs, we utilized an anti-human layilin-crosslinking antibody (*Mahuron et al., 2020*). Layilin-expressing human Tregs were treated with anti-layilin-crosslinking antibody, and LFA-1 activation was quantified by flow cytometry using the m-24 antibody binding, a monoclonal antibody (mAb) that especially recognizes LFA-1 in its open, active conformation (*Dransfield and Hogg, 1989*). These experiments were performed with and without anti-CD3/anti-CD28 stimulation to determine whether TCR stimulation can enhance the effect of layilin on LFA-1 activation. Both in non-stimulated and stimulated conditions, crosslinking layilin on Tregs resulted in increased LFA-1 activation, as measured by increased m-24 mAb staining (*Figure 4D*).

To bolster our findings with layilin crosslinking antibodies, we set out to repeat these experiments with a natural ligand for layilin. Layilin has been shown to bind to collagen IV (*Glasgow et al., 2022*). Thus, to confirm layilin's impact on LFA-1 activation, we used our CRISPR system to reduce layilin expression (*Figure 4—figure supplement 1*) and plated the cells from five different donors on non-coated or collagen IV-coated plates. Consistent with the results using layilin crosslinking mAbs, these experiments revealed that CRISPR-mediated *Layn* editing reduces LFA-1 activation only in the presence of collagen IV (*Figure 4E*). Finally, we tested the direct adhesive capacity of Tregs using an adhesion assay under the same coated plate conditions. Cells were sort-purified on layilin expression (as described above) to obtain pure populations of layilin-expressing and layilin-deficient Tregs. Consistent with our previous findings, Tregs deficient in layilin displayed a reduced capacity to bind to collagen IV-coated plates but not uncoated plates (*Figure 4F*). Taken together, this data reveals that engagement of layilin on Tregs results in enhanced LFA-1 activation and cellular adhesion.

## Layilin induces cytoskeleton changes in Tregs

Actin polymerization is one of the major cytoskeleton changes involved in cell migration and adhesion (*Gardel et al., 2010*). Thus, we hypothesized that signaling through layilin would induce cytoskeleton changes consistent with increased adhesion. To test this, we measured actin dynamics in cells by performing F-actin staining on Tregs sort-purified based on layilin expression after CRISPR-editing with *Layn*-specific or control guide RNAs (as described in *Figure 4A*). These experiments were performed in the presence of layilin ligand (collagen IV) or irrelevant ligand (fibronectin). These experiments revealed increased F-actin staining and increased cellular projections (consistent with lamellipodia) in

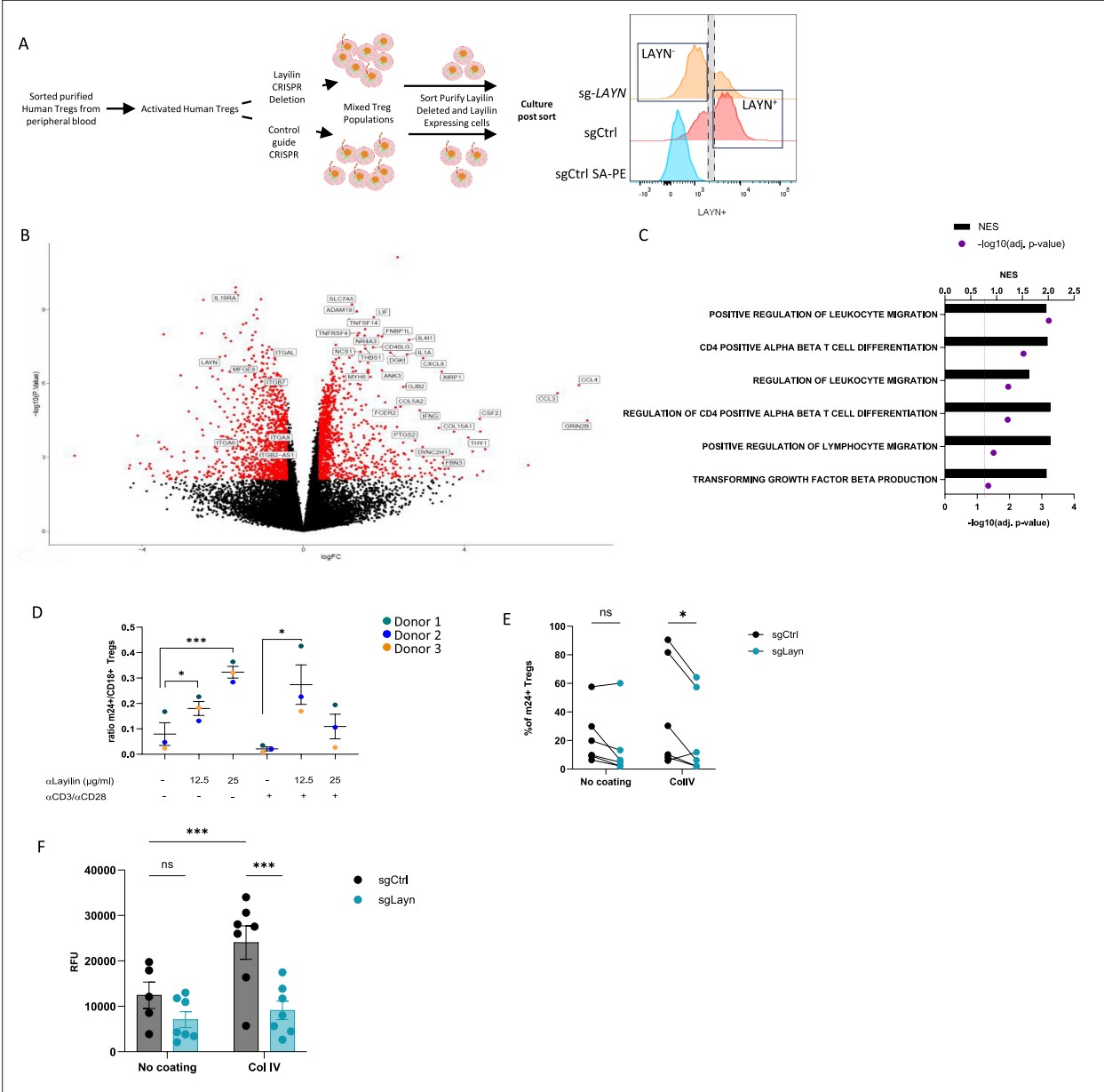

**Figure 4.** Layilin enhances regulatory T cell (Treg) adhesion and lymphocyte function-associated antigen-1 (LFA-1) activation. (**A**) The experimental design strategy starts with the isolation of human Tregs from peripheral blood, which were activated and expanded using anti-CD3/anti-CD28 stimulation and IL-2 for 12 days. We deleted *LAYN* using a CRISPR/Cas9; after 4 days, cells were sorted by flow cytometry based on LAYN expression, as represented by the representative histogram of LAYN expression on sgCtrl (stained with or without primary antibody detecting layilin followed by streptavidin-PE staining) or sg-*LAYN* Tregs, to obtain a pure population of LAYN+ or LAYN- Tregs. After sorting, cells were placed in culture for follow-up experiments presented in panels (**B**), (**C**), and (**F**). (**B**) Volcano plot of differential genes expressed in LAYN+ (on the left) or LAYN- (on the right) Tregs put in culture for 24 hours in the presence of anti-CD3/anti-CD28. Red dots represent differentially expressed genes, and black dots are not differentially expressed ones. n=4 paired donors for each group. (**C**) Top deregulated pathways obtained from a gene set enrichment analysis comparing LAYN+ to LAYN- Tregs. The top axis and bar plot represent the normalized enrichment score (NES). The bottom axis and dot represent the adjusted p-value associated with pathways enrichment. n=4 paired donors for each group. (**D**) Ratio of m24+ to total CD18+ Tregs gated on live CD8-CD4+CD25+FOXP3+ cells after 3 days culture with or without anti-CD3/anti-CD28 stimulation and in the presence of a layilin-crosslinking antibody as described on the x-axis for 20 minutes during LFA-1 activation staining. n=3 donors. Significance was defined by one-way ANOVA, followed by Dunnett's multiple-comparisons test. *p<0.05, ***p<0.001. (**E**) Frequency of m24+ sg-*LAYN* and sg-CTRL Tregs gated on live CD8-CD4+CD25+FOXP3+ cells after 24 hours of culture in a non-coated plate or collagen IV-coated plate. n=6 donors from two independent experiments. Significance was defined by two-way ANOVA, followed by Sidak's multiple-comparisons test. *p<0.05. (**F**) Numbers of cells attached to the bottom of the well are measured by the cell titer blue (expressed by relative fluorescence unit). n=7 donors from two independent experiments. Significance was defined by two-way ANOVA. ns, nonsignificant, ***p<0.001.

*Figure 4 continued on next page*

*Figure 4 continued*

The online version of this article includes the following figure supplement(s) for figure 4:

**Figure supplement 1.** Characterization of human ex vivo experiments.

layilin-deficient cells only in the presence of collagen IV (***Figure 5A and B***). As previously shown, layilin contains a talin binding domain, which we have shown to be essential for layilin-LFA-1 interaction in exhausted CD8[+] T cells (***Mahuron et al., 2020***). Furthermore, talin, vinculin, and paxillin are recruited to form a complex and interact with the binding domain of high-affinity conformation integrins, leading to focal adhesion and slowing cells (***Jankowska et al., 2018***). Conversely, an increase in polymerized actin content and cellular protrusions correlates with a higher motile profile (***Dupré et al., 2015***; ***Yan et al., 2019***). Lamellipodia are the main protrusive structures formed by T cells during migration and require actin-related protein 3 (ARP3) (***Obeidy et al., 2020***). Taken together, we propose a working model (***Figure 5C***) in which layilin, by its interaction with LFA-1 and talin, stabilizes the formation of focal adhesion complexes and favors a more anchored Treg cell phenotype. Conversely, the absence of layilin promotes cell motility, at least in part, by enhancing lamellipodia formation.

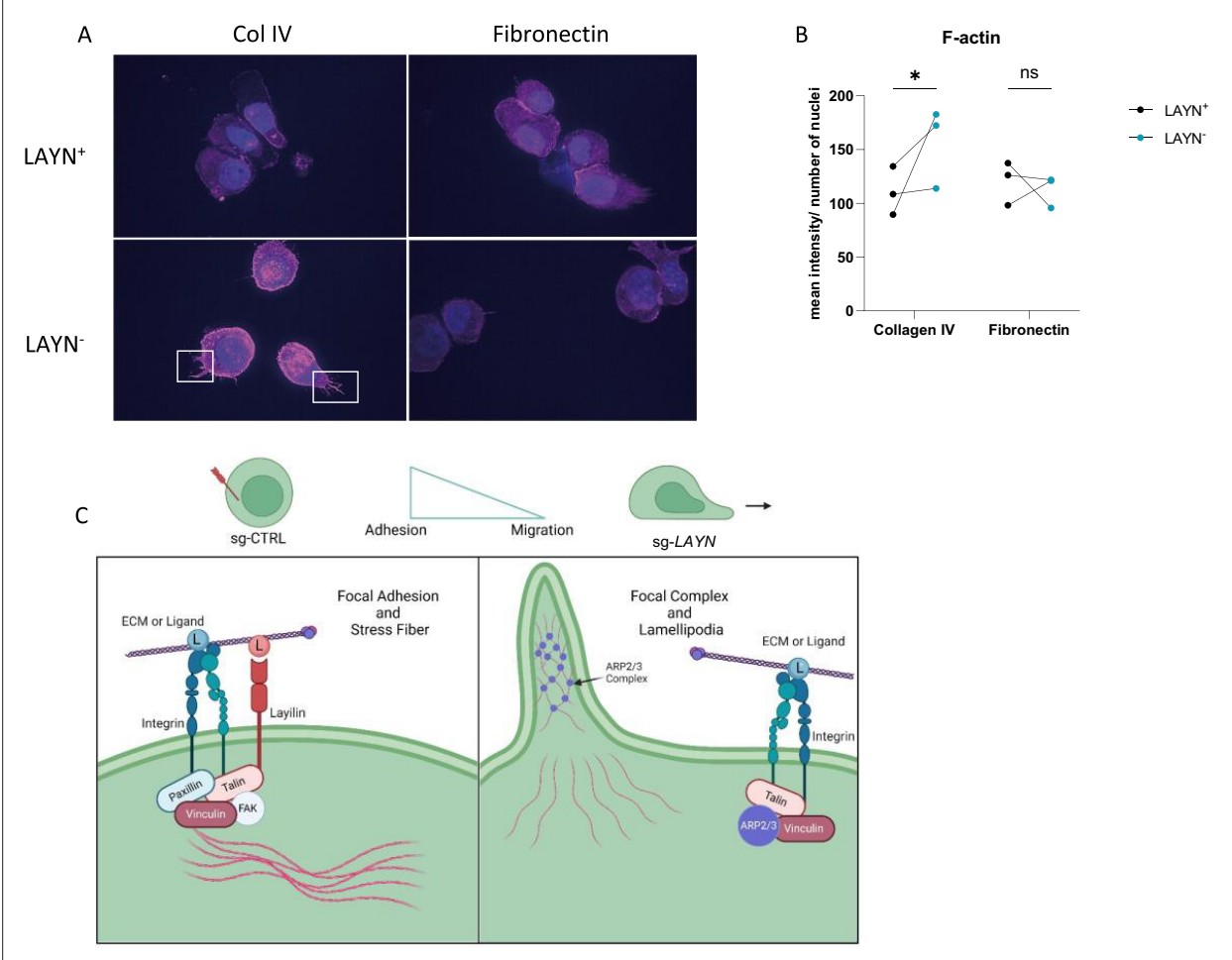

**Figure 5.** Layilin induces cytoskeleton changes in regulatory T cells (Tregs) indicative of reduced motility. (**A**) Representative pictures of LAYN[+] (on the top) or LAYN[-] (on the bottom), Tregs phalloidin staining (in pink) and nuclei (in blue) after a 48-hour culture on collagen IV (left) or fibronectin (right)-coated plates. White boxes highlight protuberances at the membranes. (**B**) Quantification of phalloidin staining associated with pictures in (**A**). n=3 donors; 10 pictures were quantified for each donor, and experiments were repeated independently twice. Significance was defined by two-way ANOVA, followed by Sidak's multiple-comparisons test. *p<0.05. (**C**) Graphical hypotheses of layilin pathway mechanism of action. In the presence of layilin, it interacts with lymphocyte function-associated antigen-1 (LFA-1) and maybe other integrins and stabilizes the formation of a focal adhesion complex, leading to the anchoring mechanism of the cells. Without layilin, integrins are less prone to adhesion mechanisms, and the ARP3 complex may act in lamellipodia formation, leading to more migratory Tregs.

## Discussion

In this study, we discovered that layilin, a C-type lectin cell surface receptor, is highly expressed by Tregs in skin. The summation of our data from results presented herein and previous studies (*Mehta et al., 2021*) suggests that layilin modulates Treg motility in tissues. This effect on motility is mediated, at least in part, through the modulation of LFA-1 activation and contributes to cytoskeletal remodeling necessary for cellular adhesion and movement.

Our data supports a model whereby layilin expression is induced on Tregs upon entering non-lymphoid tissues such as skin. This acts to anchor Tregs in the tissue, possibly adjacent to (or within) collagen IV-rich blood vessels (*Uitto et al., 1989*; *Pfisterer et al., 2021*). In this capacity, layilin may help sequester Tregs to a defined niche, facilitating their ability to maintain tissue immune homeostasis in the steady state. Conversely, in pathological situations such as tumors and chronic inflammation, the motility of Tregs may play a larger role in their functionality, allowing them to navigate to various tissue microanatomic niches to attenuate specific cell types and inflammatory mediators. Notably, several studies have underscored the crucial role of adhesion properties, particularly those mediated by ICAM-1, the main adhesion molecule expressed by T cells, in enabling effective Treg suppression capacity (*Gottrand et al., 2015*; *Tran et al., 2009*; *Chen et al., 2017*). In this context, layilin expression on Tregs most likely acts to curtail their suppressive capacity by limiting their motility. Indeed, two-photon microscopy studies revealed that layilin-deficient Tregs are relatively more motile in inflamed skin compared to layilin-expressing control Tregs (*Mehta et al., 2021*). In support of this theory, herein, we show that reducing layilin expression on Tregs results in enhanced immune regulation in a skin inflammation model, as demonstrated by a significant reduction in pro-inflammatory cells, observed in *Foxp3^Cre^/Layn^fl/fl^* mice compared to controls. In contrast, it is interesting to highlight that, in vitro, layilin plays a minimal role in Treg activation and suppressive capacity in vitro. This suggests the role of layilin in the suppression capacities of Tregs depends on the more complex, dynamic, and collagen-rich in vivo environment.

An intriguing finding is the association between layilin and LFA-1 activation observed in Tregs, echoing previous discoveries in CD8+ T cells (*Mahuron et al., 2020*). LFA-1 is crucial for T cell engagement at the immune synapse (*Hammer et al., 2019*; *Franciszkiewicz et al., 2013*). While the modulation of LFA-1 by layilin may enhance Treg interactions with other cells, potentially boosting their suppressive abilities, it could also confine their impact to a narrowly defined area due to reduced motility.

Layilin's role in enhancing the effector functions of CD8+ T cells in tumors (*Zheng et al., 2017*; *Mahuron et al., 2020*; *Pan et al., 2019*) and limiting Treg-suppressive capacity in inflamed tissues suggests that it could be a valuable therapeutic target for autoimmune and chronic inflammatory diseases. A layilin neutralizing antibody could attenuate CD8+ T cell killing and enhance Treg suppression, effectively working in concert to dampen tissue inflammation. Tissue-specific autoimmune diseases thought to be mediated by autoreactive CD8+ T cells and suboptimally controlled by Tregs would be attractive indications. However, more research is necessary to understand the role of layilin on non-immune cells, such as gastrointestinal epithelial cells (*Bono et al., 2001*; *Kim et al., 2018*), to help build confidence around this therapeutic hypothesis.

## Materials and methods

### Experimental animals

*Foxp3^GFPCreERT2^* mice were purchased from the Jackson Laboratory (ref #016961, Bar Harbor, ME) and were bred and maintained at the University of California San Francisco (UCSF) specific pathogen-free facility. As previously described by our group, *Layn^fl/fl^* mice were created by inserting LoxP sites flanking exon 4 of the *Layn* gene using CRISPR-Cas9 (*Mehta et al., 2021*). Layilin was deleted specifically on Tregs by crossing *Layn^fl/fl^* mice to *Foxp3^GFPCreERT2^* mice upon treatment with tamoxifen. All mouse experiments were performed on 7–12-week-old animals. All mice were housed under a 12-hour light/dark cycle. All animal experiments were performed following the guidelines established by the Laboratory Animal Resource Center at UCSF, and all experimental plans and protocols were approved by the Institutional Animal Care and Use Committee (IACUC) beforehand.

### Mouse tissue processing

Briefly, mouse skin was minced and digested in a buffer containing collagenase XI, DNase, and hyaluronidase in complete Roswell Park Memorial Institute (RPMI) medium in an incubator shaker at

225 rpm for 45 minutes at 37°C. Once the single-cell suspensions were obtained, cells were counted using an automated cell counter (NucleoCounter NC-200, Chemometec). 2–4 × 10$^6$ cells were stained, and flow cytometric analysis was performed.

## Human specimens

Normal healthy human skin was obtained from patients at the UCSF undergoing elective surgery, and the specimens received were de-identified (study number 19-29608) and certified as Not Human Subjects Research. Blood samples were obtained from healthy adult volunteers (study number 12-09489) or ordered from the Standford Blood Center. Biopsies of psoriasis were obtained with a 6 mm punch biopsy tool (study number 19-29608). For all fresh human tissue samples, active/flaring disease was confirmed both clinically and histologically. Studies using human samples were approved by the UCSF Committee on Human Research and the institutional review board of UCSF. Informed written consent was obtained from all patients.

## Human skin digestion

Skin samples were stored in a sterile container on gauze and PBS at 4°C until digestion. Skin was processed and digested as previously described. Briefly, hair and subcutaneous fat were removed, and skin was cut into small pieces and mixed with digestion buffer containing 0.8 mg/ml collagenase type 4 (4188; Worthington), 0.02 mg/ml DNAse (DN25-1G; Sigma-Aldrich), 10% Fetal bovine serum (FBS), 1% 4-(2-hydroxyethyl)-1-piperazineethanesulfonic acid (HEPES), and 1% penicillin/streptavidin in RPMI medium and digested overnight in an incubator. They were washed (2% FBS, 1% penicillin/streptavidin in RPMI medium), double filtered through a 100 μm filter, and cells were pelleted and counted.

## Tregs are derived from human peripheral blood

CD4$^+$ T cells were purified using RosetteSep (Stem Cell), followed by a gradient of Ficoll-Paque Plus (GE Healthcare) from the peripheral blood of healthy volunteers. Once isolated, CD4$^+$ T cells were stained and sorted by gating on live CD4$^+$CD8$^-$CD25$^{hi}$CD127$^{lo}$ cells. Purified Tregs were ex vivo expanded for 12 days in complete X-Vivo media (Lonza) with 500 U/ml IL-2 (2000 U/ml, Tonbo Biosciences) and stimulated with anti-CD3/CD28 beads at a cells:beads ratio of 1:1 (Gibco). At D12, cells were frozen for later experiments.

## Human Tregs engineering/CRISPR

To engineer human Treg, we used a CRISPR-cas9 SNP protocol. We thawed D12 Tregs and rested them for 4 hours. Afterward, we mixed 22.2 pmol of *Streptococcus pyogenes* Cas9 with 88.8 pmol of scramble single guide (sg-CTRL) or *LAYN* single guide (sg-*LAYN*, UUCUCGAAGACUGAACUUUG) and incubated for 10 minutes at room temperature to form the RNP complex. During the incubation, we harvested the cells, counted them using the NucleoCounter, and resuspended them in P3 buffer cells at a 2 million/20 ul concentration. Once the RNP complex was formed, an electroporation enhancer (IDT) was added at a ratio of 1:2 (EE:RNP) to the cells in an electroporation plate. Cells were electroporated using the DG-113 program on a 4D-Nucleofector 96-well unit (Lonza). After 4 days, efficiency was measured, and cells were used for the experiment.

## Single-cell RNA sequencing

Sequencing data are available on the National Center for Biotechnology Information Gene Expression Omnibus (GEO): GSE249622, GSE249793, and GSE283500. scRNA-seq and scTCR-seq libraries were prepared at the UCSF Core Immunology laboratory using the 10X Chromium Single Cell 5′ Gene Expression and V(D)J Profiling Solution kit, according to the manufacturer's instructions (10X Genomics). 150 paired-end sequencing was performed on a NovaSeq 6000 instrument. The Cell Ranger analysis pipelines were then used to process the generated sequencing data. Data were demultiplexed into FASTQ files, aligned to the GRCh38 human reference genome, and counted, and TCR library reads were assembled into single-cell V(D)J sequences and annotations. For gene expression analysis, the R package Seurat was used. Filtered gene-barcode matrices were loaded, and quality control steps were performed (low-quality or dying cells and cell doublets/multiplets were excluded from subsequent analysis). Data were normalized and scaled, then linear dimensional reduction using

principal component analysis was performed. Highly variable genes were used to perform unsupervised clustering, and non-linear dimensional reduction with Uniform Manifold Approximation and Projection (UMAP) was used to visualize the data.

For differential gene expression analysis comparing *LAYN*-expressing cells to *LAYN*-non-expressing cells, data were subsetted into *LAYN*-negative (*LAYN*<0) and *LAYN*-positive (*LAYN*>0) using *LAYN* expression >0 as a threshold. For TCR analysis, samples were downsampled to get the same number of cells for each donor. All plots were generated using the R packages ggplot2 and cowplot.

## Conventional and spectral flow cytometry

Single-cell suspensions were counted and pelleted. Cells were washed and stained with Ghost Viability dye (Tonbo Biosciences) and antibodies against surface markers in PBS. Cells were fixed and permeabilized for intracellular staining using a FoxP3 staining kit (eBiosciences) and then stained with antibodies against intracellular markers. Fluorophore-conjugated antibodies specific for human or mouse surface and intracellular antigens were purchased from SinoBiological, Cytek, BD Biosciences, eBiosciences, or BioLegend. The following anti-human antibodies and clones were used: layilin (clone #07, Sino Biological Cat# 10208-MM07, RRID:AB_2860113), PE-Streptavidin, CD3 (UCHT1), CD4 (SK3), CD8 (SK1), CD45RO (UCHL1), FoxP3 (PCH101), CD25 (M-A251), CTLA4 (14D3), ICOS (ISA-3), CD27 (O323), CD11c (3.9), HLA-DR (L243), CD137 (4B4-1), PD1 (EH12.2H7), Ki67 (EH12.2H7), LAG3 (11C3C65), CD38 (HIT2), CD62L (DREG56), CD69 (FN50), CD127 (eBioRDR5), GATA3 (L50-823), GZMB (GB11), and HELIOS (22F6). Layilin Ab was biotin-tagged using Miltenyi's One-Step Antibody Biotinylation kit (130-093-385). The following anti-mouse antibodies and clones were used: CD3 (145-2C11), CD4 (RM4-5), CD8 (53-6.7), CD45 (30-F11), FoxP3 (FJK-16s), TCRb (H57-597), CD25 (PC61.5), CTLA4 (UC10-4B9), ICOS (C398.4A), Ki67 (B56), CD44 (IM7), CD69 (H1.213), IL-17 (TC11-18H10.1), IFNγ (XMG1.2), and TNFα (MP6-XT22). Samples were run on a Fortessa analyzer (BD Biosciences) for conventional flow cytometry or an Aurora analyzer (Cytek) for spectral flow cytometry in the UCSF Flow Cytometry Core. Data was collected using FACS Diva software (BD Biosciences). Data were analyzed using FlowJo software (FlowJo, LLC). Dead cells and doublet cell populations were excluded, followed by pre-gating on CD45$^+$ populations for immune cell analysis. Lymphoid cells were gated as TCRβ$^+$CD3$^+$ T cells, CD3$^+$CD8$^+$ T cells (CD8), CD3$^+$CD4$^+$CD25$^-$Foxp3$^-$ T effector cells (Teff), and CD3$^+$CD4$^+$CD25$^+$Foxp3$^+$ Tregs. For human Tregs, expression of layilin was quantified based on isotype control antibody or secondary control.

## RNA-sequencing analysis

For human Tregs, cells were sorted based on layilin expression and put in culture for 24 hours in the presence of anti-CD3/anti-CD28 stimulation. After 24 hours, cells have been harvested and spun down to obtain a pellet, followed by a flash freezing of the subsequent. Samples have been shipped to Novogene for RNA extraction, library preparation, and sequencing.

For the IMQ mouse model, Treg and Teff cells were isolated respectively by gating on live CD45$^+$CD3$^+$CD4$^+$CD8$^-$CD25$^{hi}$CD62L$^{hi}$GFP$^+$ and CD45$^+$CD3$^+$CD4$^+$CD8$^-$CD25$^{lo}$CD62L$^{hi}$GFP$^-$ cells. Cells were sorted into lysis buffer from the kit and frozen on dry ice before being processed using the SMART-Seq v4 Ultra Low Input RNA Kit (Takara Bio) according to the manufacturer's protocol. 100 pg of amplified cDNA obtained was used to generate a library using Nextera XT DNA preparation kit (Illumina). The quality of cDNA and library products was checked and quantified using an Agilent High Sensitivity DNA chip. An equal quantity of libraries was indexed and pooled together before being sent to Novogene.

We used NovaSeq 6000 PE150 to a 30M read depth for all sequencing. Reads were aligned to Ensembl GRCm39 (for mouse) or GRCh38 (for human) reference genome using Kallisto (RRID:SCR_016582). The R/Bioconductor package Limma voom was used to determine differential expression (*Law et al., 2014*; *Ritchie et al., 2015*). Sequencing data are available on the National Center for Biotechnology Information GEO with accession numbers GSE282563 (for human) and GSE282465.

## In vitro human Treg suppression assay

Before plating, the responder cells (CD4$^+$ T effectors and CD8$^+$ T cells) were stained with cell trace violet (CTV, Thermo Fisher) at 7 million/ml concentration and CTV diluted at 1:8000 in PBS. After a 30-minute incubation in the dark at 37°C, cells were washed and resuspended at a concentration of

$5 \times 10^5$ cells/ml. 50K responder cells were plated in 96-well U-bottom plates, and appropriate cell numbers equivalent to each ratio of Tregs CRISPR were added to the plate, as well as anti-CD3/anti-CD28 stimulation. After a 3-day incubation, the proliferation of responder cells was measured using CTV quantification and flow cytometry.

## Imiquimod model

Mice were treated for 5 days with tamoxifen to induce Cre recombinase before the start of the treatment. On the last day of tamoxifen treatment, the back skin of the mice was shaved. Two days later, we measured the clinical score of mice to establish a baseline with the following readouts: erythema, scaling, skin lesions, and weight. We then started the IMQ 5% cream (Taro) application for six consecutive days. On day 7 (for flow cytometry) or day 5 (for bulk RNAseq), mice were sacrificed, and skin and sdLN were harvested. Samples were stained and analyzed by flow cytometry.

## LFA-1 activation assay

100,000 cells were plated in a 96-well U-plate for 24 hours in combination with appropriate treatment. To report LFA-1 activation, cells were stained at 37°C with clone m24 (BioLegend) in an affinity buffer containing 20 mM HEPES, 140 mM NaCl, 1 mM $MgCl_2$, 1 mM $CaCl_2$, 2 mg/ml glucose, and 0.5% bovine serum albumin (BSA). 2 mM $MnCl_2$ was used as a positive control and a combination of 2 mM $MnCl_2$ + 2 mM EDTA as negative control. For layilin crosslinking, layilin antibody was added simultaneously using the following concentrations: 12.5 or 25 ug/ml (Sino Biological).

## In vitro human Treg adhesion assay

The day before, the plate was coated with 2.5 ug/ml of collagen IV (C5533, MilliporeSigma) and incubated overnight at 4°. Cells were sorted based on layilin expression and put in culture for 24 hours in the presence of anti-CD3/anti-CD28 stimulation. After incubation, the plate was washed twice by flipping it to remove any unbound cells. To measure the cells attached, we used Cell Titer Blue (Promega) and read the fluorescence using a plate reader.

## F-actin staining

Cells were sorted based on layilin expression and put in culture for 48 hours in the presence of anti-CD3/anti-CD28 stimulation. After 48 hours, cells were harvested, and a cytospin was performed to transfer cells on a slide. After being fixed for 10 minutes at room temperature using 4% paraformaldehyde (PFA), slides were washed and stained with phalloidin Alexa Fluor 546 (Invitrogen) diluted in a mix of PBS containing 1% BSA and 0.1% saponin for an hour at room temperature protected from light. After washing in PBS, slides were mounted with a hard mounting solution containing DAPI (Southern Biotech). Images ×100 were obtained using a confocal microscope.

## Quantitative PCR

For the assessment of *Layn* gene expression, Tregs and Teffs were sort-purified from skin and sdLNs of *Foxp3$^{CreERT2}$; Layn$^{fl/fl}$* and *Foxp3$^{CreERT2}$* mice. RNA was isolated using a column-based kit (PureLink RNA Mini Kit, Thermo Fisher), and then transcribed (iScript cDNA Synthesis Kit, Bio-Rad). Expression of *Layn* was determined using a SYBR Green assay (SSo Advanced Universal SYBR Green kit; Bio-Rad). The cycle number of duplicate or triplicate samples was normalized to the expression of the endogenous control Rsp17. Primer sequences used are for Rsp17: for: 5'-CGCCATTATCCCCAGCAAG-3'; rev 5'-TGTCGGGATCCACCTCAATG-3'; for layn: for: 5'-TCCATGACGCCTTTCAAAGAC-3'; rev 5'-AGGCTGTGTTATTGCTCTGTTTC-3'. Data are presented as relative arbitrary units (AU).

## Statistical analyses

Statistical analyses were performed with Prism software package version 10 (GraphPad). p-Values were calculated using two-tailed unpaired or paired Student's *t*-test for comparison between two groups, one- or two-way ANOVA, followed by the appropriate post hoc test for comparison including more than two groups and/or more than one condition. Pilot experiments were used to determine the sample size for animal experiments. No animals were excluded from the analysis unless there were technical errors. Mice were age- and gender-matched and randomly assigned into experimental groups. Appropriate statistical analyses were applied, assuming a normal sample distribution. All in

vivo mouse experiments were conducted with at least 2–3 independent animal cohorts. RNA-seq experiments were conducted using 4–5 biological samples (as indicated in figure legends). Data are mean ± S.E.M. p-Values correlate with symbols as follows: ns = not significant, p>0.05, *p<0.05, **p<0.01, ***p<0.001, ****p<0.0001.

## Acknowledgements

We thank the patient donors for providing tissue samples for this study. We also thank Clare Abram and Cuyler Luck for assistance with western blotting. Flow cytometry data were generated in the UCSF Parnassus Flow CoLab (RRID:SCR_018206), supported in part by grant NIH P30 DK063720 and by the NIH S10 Instrumentation grant S10 1S10OD026940-01 and S10 1S10OD021822-0. This work was primarily supported by TRex Bio Inc.

## Additional information

### Competing interests

Jose Valle, Mariela Pauli, Hong-An Truong, Karin Reif, Ali A Zarrin: employee of TRex Bio Inc. Jarish Cohen: consultant for TRex Bio Inc and Radera Bio Inc. Maxime Kinet, Margaret M Lowe: consultant for Radera Bio Inc. Michael D Rosenblum: consultant and co-founder of TRex Bio Inc, Sitryx Bio Inc, and Radera Bio Inc; also a consultant for Mozart Bio Inc. The other authors declare that no competing interests exist.

### Funding

| Funder | Grant reference number | Author |
|---|---|---|
| National Institute of Arthritis and Musculoskeletal and Skin Diseases | 5R21AR072195-02 | Victoire Gouirand<br>Sean Clancy<br>Courtney Macon<br>Jose Valle<br>Mariela Pauli<br>Hong-An Truong<br>Jarish Cohen<br>Maxime Kinet<br>Margaret M Lowe<br>Samuel J Lord<br>Kristen Skruber<br>Hobart Harris<br>Esther Kim<br>Isaac Neuhaus<br>Karin Reif<br>Ali A Zarrin<br>R Dyche Mullins<br>Michael D Rosenblum |
| TRex Bio Inc | | Jose Valle<br>Mariela Pauli<br>Hong-An Truong<br>Karin Reif<br>Ali A Zarrin |

The funders had no role in study design, data collection and interpretation, or the decision to submit the work for publication.

### Author contributions

Victoire Gouirand, Conceptualization, Data curation, Formal analysis, Supervision, Investigation, Methodology, Writing – original draft, Project administration, Writing – review and editing; Sean Clancy, Samuel J Lord, Kristen Skruber, Formal analysis; Courtney Macon, Jose Valle, Mariela Pauli, Hong-An Truong, Jarish Cohen, Maxime Kinet, Methodology; Margaret M Lowe, Investigation; Hobart Harris, Esther Kim, Isaac Neuhaus, Resources, Investigation; Karin Reif, Data curation, Formal analysis, Investigation; Ali A Zarrin, Resources, Investigation, Methodology; R Dyche Mullins, Investigation, Methodology; Michael D Rosenblum, Conceptualization, Data curation, Formal analysis, Supervision,

Funding acquisition, Investigation, Writing – original draft, Project administration, Writing – review and editing

**Author ORCIDs**
Victoire Gouirand (ID) https://orcid.org/0000-0002-2666-0061
Samuel J Lord (ID) https://orcid.org/0000-0002-2785-989X
Karin Reif (ID) https://orcid.org/0000-0001-5490-8389
R Dyche Mullins (ID) https://orcid.org/0000-0002-0871-5479
Michael D Rosenblum (ID) https://orcid.org/0000-0002-0462-5732

**Ethics**

Human subjects: Normal healthy human skin was obtained from patients at UCSF undergoing elective surgery, and the specimens received were deidentified (study number 19-29608) and certified as Not Human Subjects Research. Blood samples were obtained from healthy adult volunteers (study number 12-14 09489) or ordered from the Standford Blood Center. Biopsies of psoriasis were obtained with a 6-mm punch biopsy tool (study number 19-29608). For all fresh human tissue samples, active/flaring disease was confirmed both clinically and histologically. Studies using human samples were approved by the UCSF Committee on Human Research and the institutional review board of UCSF. Informed written consent was obtained from all patients.

All animal experiments were performed following guidelines established by the Laboratory Animal Resource Center at UCSF, and all experimental plans and protocols were approved by IACUC before-hand (AN110246-01C).

Reviewer #1 (Public review): https://doi.org/10.7554/eLife.105277.2.sa1
Reviewer #2 (Public review): https://doi.org/10.7554/eLife.105277.2.sa2
Reviewer #3 (Public review): https://doi.org/10.7554/eLife.105277.2.sa3
Author response https://doi.org/10.7554/eLife.105277.2.sa4

---

# Additional files

**Supplementary files**
MDAR checklist
Source code 1. R analysis scripts used in this study.

**Data availability**

scRNAseq data are available on GEO with accession numbers GSE249622, GSE249793, and GSE283500. Bulk RNAseq data are available on GEO with accession numbers GSE282563 (for human) and GSE282465 (for mice). Coding scripts have been provided as *Source code 1*. All correspondence and requests for materials can be made to the corresponding author, MDR. All other data needed to evaluate the conclusions of this study are present in the paper.

The following datasets were generated:

| Author(s) | Year | Dataset title | Dataset URL | Database and Identifier |
|---|---|---|---|---|
| Lowe MM, Cohen JN, Moss MI, Clancy S | 2023 | scRNASeq Data of CD4 Tregs and Teffectors in HS lesional skin and healthy skin | https://www.ncbi.nlm.nih.gov/geo/query/acc.cgi?acc=GSE249622 | NCBI Gene Expression Omnibus, GSE249622 |
| Lowe MM, Cohen JN, Moss MI, Clancy S | 2023 | scRNASeq Data of CD3 Negative Immune Cells in HS lesional skin, healthy skin, and matched blood | https://www.ncbi.nlm.nih.gov/geo/query/acc.cgi?acc=GSE249793 | NCBI Gene Expression Omnibus, GSE249793 |
| Rosenblum MD | 2025 | Layilin Regulates Treg Motility and Suppressive Capacity in Skin | https://www.ncbi.nlm.nih.gov/geo/query/acc.cgi?acc=GSE283500 | NCBI Gene Expression Omnibus, GSE283500 |

*Continued on next page*

*Continued*

| Author(s) | Year | Dataset title | Dataset URL | Database and Identifier |
|---|---|---|---|---|
| Rosenblum MD | 2025 | Layilin Regulates Treg Motility and Suppressive Capacity in Skin | https://www.ncbi.nlm.nih.gov/geo/query/acc.cgi?acc=GSE282563 | NCBI Gene Expression Omnibus, GSE282563 |
| Rosenblum MD | 2025 | Layilin Regulates Treg Motility and Suppressive Capacity in Skin | https://www.ncbi.nlm.nih.gov/geo/query/acc.cgi?acc=GSE282465 | NCBI Gene Expression Omnibus, GSE282465 |

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
