## [Editor Report · eLife Assessment]

This study reports **valuable** findings on the role of layilin in the motility and suppressive capacity of clonal expanded regulatory T cells (Tregs) in the skin. Although the strength of the study is utilizing conditional knock-out mice and human skin samples, the analysis of the molecular mechanism by which layilin affects Treg function is **incomplete**. The study will be of interest to medical scientists working on skin immunology.

---

## [Referee Report · Reviewer #1 (Public review)]

Summary and Strengths:

This work shows that the gene encoding Layilin is expressed preferentially in human skin Tregs, and that the fraction of Tregs expressing Layilin may overexpress genes related to T cell activation and adhesion. Expression of Layilin on Tregs would have no impact on activation markers or in vitro suppressive function. However, activation of Layilin either with a cross-linking antibody or collagen IV, its natural ligand, would promote cell adhesion via LFA1 activation. The in vivo functional role of Layilin in Tregs is studied in a conditional KO mouse model in a model of skin inflammation. Deletion of Layilin in Tregs led to an attenuation of the disease score and a reduction in the cutaneous lymphocyte infiltrate. This work is clearly innovative, but a number of major points limit its interest.

Weakness and major points:

(1) The number of panels and figures suggests that this story is quite complete but several data presented in the main figures do not provide essential information for a proper understanding of Layilin's role in Tregs.

Figures 1I, 1J, and the whole of Figure 2 could be placed as supplementary figures. Also, for Figure 3E, it would be preferable to show the percentage of cells expressing cytokines rather than their absolute numbers. In fact, the drop in the numbers of cytokine-producing cells is probably due solely to the drop in total cell numbers and not to a decrease in the proportion of cells expressing cytokines. If this is the case, these data should be shown in supplementary figures. Finally, Figures 4 and 5 could be merged.

(2) Some important data are not shown or not mentioned.

(a) It would be important to show the proportion of Treg, Tconv, and CD8 expressing Layilin in healthy skin and in patients developing psoriasis, as well as in the blood of healthy subjects.

(b) We lack information to be convinced that there is enrichment for migration and adhesion genes in Layilin+ Tregs in the GSEA data. The authors should indicate what geneset libraries they used. Indeed, it is tempting to show only the genesets that give results in line with the message you want to get across. If these genesets come from public banks, the bank used should be indicated, and the results of all gene sets shown in an unbiased way. In addition, it should be indicated whether the analyses were performed on untransformed or pseudobulk scRNAseq data analyses. Finally, it would be preferable to confirm the GSEA data with z-score analyses, as Ingenuity does, for example. Indeed, in GSEA-type analyses, there are genes that have activating but also inhibiting effects on a pathway in a given gene set.

(c) For all FACS data, the raw data should be shown as histograms or dot plots for representative samples.

(d) For Figure 5B, the number of samples analyzed is insufficient to draw clear conclusions.

(3) For Figs. 4 and 5, the design of the experiment poses a problem. Indeed, the comparison between Layn+ and Layn- cells may, in part, not be directly linked to the expression or absence of expression of this protein. Indeed, Layn+ and Layn- Tregs may constitute populations with different biological properties, beyond the expression of Layn. However, in the experiment design used here, a significant fraction of the sorted Layn- Tregs will be cells belonging to the population that has never expressed this protein. It would have been preferable to sort first the Layn+ Tregs, then knock down this protein and re-sort the Layn- Tregs and Layn+ Tregs. If this experiment is too cumbersome to perform, I agree that the authors should not do it. However, it would be important to mention the point I have just made in the text.

---

## [Referee Report · Reviewer #2 (Public review)]

Summary:

In their manuscript, Gouirand et al. report on the role of Layilin expression for the motility and suppressive capacity of regulatory T cells (Tregs). In previous studies, the authors had already demonstrated that Layilin is expressed on Tregs, that it acts as a negative regulator of their suppressive capacity, that it functions to anchor Tregs in non-lymphoid tissues, and that it enhances the adhesive properties of Layilin-expressing cells by co-localization with the integrin αLβ2 (LFA-1). Building on these published data, the authors now show that Layilin is highly expressed on a subset of clonally expanded effector Tregs in both healthy and psoriatic skin and that deletion of Layilin in Tregs in vivo resulted in significantly attenuated skin inflammation. Furthermore, the authors addressed the molecular mechanism by which Layilin affects the suppressive capacity of Tregs and showed that Layilin increased Treg adhesion via modulation of LFA-1, resulting in distinct cytoskeletal changes.

Strengths:

Certainly, the strength of this study lies in the combination of data from mouse and human models.

Weaknesses:

Some of the conclusions drawn by the authors must be treated with caution, as the experimental conditions were not always appropriate, leading to a risk of misinterpretation.

---

## [Referee Report · Reviewer #3 (Public review)]

Summary:

Gouirand et al explore the function of Layilin on Treg in the context of psoriasis using both patient samples and a conditional mutant mouse model. They perform functional analysis in the patient samples using Cas9-mediated deletion. The authors suggest that Layilin works in concert with integrins to bind collagen IV to attenuate cell movement.

The work is well done and built on solid human data. The report is a modest advance from the authors' previous report in 2021 that focused on tumor responses, with this report focusing on psoriasis. There are some experimental concerns that should be considered.

Strengths:

(1) Good complementation of patient and animal model data.

(2) Solid experimentation using state-of-the-art approaches.

(3) There is clearly a biological effect of LAYN deficiency in the mouse model.

(4) The report adds some new information to what was already known from the previous reports.

Weaknesses:

(1) It is not clear that the assays used for functional analysis of the patient samples were optimal.

(2) Several conclusions are not fully substantiated.

(3) The report is lacking some experimental details.

---

## [Author Response]

**Reviewer 1:**

Concern 1: Figures 1I, 1J, and the whole of Figure 2 could be placed as supplementary figures. Also, for Figure 3E, it would be preferable to show the percentage of cells expressing cytokines rather than their absolute numbers. In fact, the drop in the numbers of cytokine-producing cells is probably due solely to the drop in total cell numbers and not to a decrease in the proportion of cells expressing cytokines. If this is the case, these data should be shown in supplementary figures. Finally, Figures 4 and 5 could be merged.

We thank you for your recommendations. As rearranging figures is not critical to convey the data, we have decided to keep the figures and supplemental figures as they are currently presented.

Concern 2a: It would be important to show the proportion of Treg, Tconv, and CD8 expressing Layilin in healthy skin and in patients developing psoriasis, as well as in the blood of healthy subjects.

This data is published in a previous manuscript from our group. Please see Figure 1 in “Layilin Anchors Regulatory T Cells in Skin” (PMID: 34470859)

Concern 2b: We lack information to be convinced that there is enrichment for migration and adhesion genes in Layilin+ Tregs in the GSEA data. The authors should indicate what geneset libraries they used. Indeed, it is tempting to show only the genesets that give results in line with the message you want to get across. If these genesets come from public banks, the bank used should be indicated, and the results of all gene sets shown in an unbiased way. In addition, it should be indicated whether the analyses were performed on untransformed or pseudobulk scRNAseq data analyses. Finally, it would be preferable to confirm the GSEA data with z-score analyses, as Ingenuity does, for example. Indeed, in GSEA-type analyses, there are genes that have activating but also inhibiting effects on a pathway in a given gene set.

Given that we have already shown that layilin plays a major role in Treg and CD8+ T cell adhesion in tissues, we used a candidate approach for our GSEA. We tested the hypothesis that adhesion and motility pathways are enriched in Layilin-expressing Tregs. There was a statistically significant enrichment for these genes in Layilin+ Tregs compared to Layilin- Tregs, which we feel adequately tests our hypothesis.

Concern 2c: For all FACS data, the raw data should be shown as histograms or dot plots for representative samples.

We respect this concern. We omit these secondary to space constraints.

Concern 2d: For Figure 5B, the number of samples analyzed is insufficient to draw clear conclusions.

We respectfully disagree. Three doners were used in a paired fashion (internally controlled) achieving statistical significance.

Concern 3: For Figs. 4 and 5, the design of the experiment poses a problem. Indeed, the comparison between Layn+ and Layn- cells may, in part, not be directly linked to the expression or absence of expression of this protein. Indeed, Layn+ and Layn- Tregs may constitute populations with different biological properties, beyond the expression of Layn. However, in the experiment design used here, a significant fraction of the sorted Layn- Tregs will be cells belonging to the population that has never expressed this protein. It would have been preferable to sort first the Layn+ Tregs, then knock down this protein and re-sort the Layn- Tregs and Layn+ Tregs. If this experiment is too cumbersome to perform, I agree that the authors should not do it. However, it would be important to mention the point I have just made in the text.

We agree. However, as the reviewer points out, these experiments are not logistically and practically feasible at this point. We do perform several experiments in this manuscript in which layilin is reduced via gene editing with results supporting our hypotheses.

**Reviewer 2:**
Some of the conclusions drawn by the authors must be treated with caution, as the experimental conditions were not always appropriate, leading to a risk of misinterpretation.

We have been transparent with all our methods and data. We will leave this to the reader to determine level of rigor and the robustness of the data.

**Reviewer 3:**
Weaknesses:It is not clear that the assays used for functional analysis of the patient samples were optimal. (2) Several conclusions are not fully substantiated. (3) The report is lacking some experimental details.

We have tried to be as comprehensive and thorough as possible. We feel that the data supports our conclusions. We will leave this to the reader to interpret and conclude.